# Neural tracking of phrases in spoken language comprehension is automatic and task-dependent

**Sanne ten Oever**[1,2,3], **Sara Carta**[1,4,5], **Greta Kaufeld**[1], **Andrea E Martin**[1,2]*

[1]Language and Computation in Neural Systems group, Max Planck Institute for Psycholinguistics, Nijmegen, Netherlands; [2]Language and Computation in Neural Systems group, Donders Centre for Cognitive Neuroimaging, Nijmegen, Netherlands; [3]Department of Cognitive Neuroscience, Faculty of Psychology and Neuroscience, Maastricht University, Maastricht, Netherlands; [4]ADAPT Centre, School of Computer Science and Statistics, University of Dublin, Trinity College, Dublin, Ireland; [5]CIMeC - Center for Mind/Brain Sciences, University of Trento, Trento, Italy

**\*For correspondence:**
Andrea.Martin@mpi.nl

**Abstract** Linguistic phrases are tracked in sentences even though there is no one-to-one acoustic phrase marker in the physical signal. This phenomenon suggests an automatic tracking of abstract linguistic structure that is endogenously generated by the brain. However, all studies investigating linguistic tracking compare conditions where either relevant information at linguistic timescales is available, or where this information is absent altogether (e.g., sentences versus word lists during passive listening). It is therefore unclear whether tracking at phrasal timescales is related to the content of language, or rather, results as a consequence of attending to the timescales that happen to match behaviourally relevant information. To investigate this question, we presented participants with sentences and word lists while recording their brain activity with magnetoencephalography (MEG). Participants performed passive, syllable, word, and word-combination tasks corresponding to attending to four different rates: one they would naturally attend to, syllable-rates, word-rates, and phrasal-rates, respectively. We replicated overall findings of stronger phrasal-rate tracking measured with mutual information for sentences compared to word lists across the classical language network. However, in the inferior frontal gyrus (IFG) we found a task effect suggesting stronger phrasal-rate tracking during the word-combination task independent of the presence of linguistic structure, as well as stronger delta-band connectivity during this task. These results suggest that extracting linguistic information at phrasal rates occurs automatically with or without the presence of an additional task, but also that IFG might be important for temporal integration across various perceptual domains.

## Editor's evaluation

This MEG study elegantly assesses human brain responses to spoken language at the syllable, word, and sentence level. Although prior studies have shown significant cortical tracking of the speech signal, the current work uses clever task manipulation to direct attention to different timescales of speech, thus demonstrating tracking mechanisms that are both automatic and task-dependent operate in tandem during spoken language comprehension.

## Introduction

Understanding spoken language likely requires a multitude of processes (*Friederici, 2011*; *Martin, 2020*; *Halle and Stevens, 1962*). Although not always an exclusively bottom-up affair, acoustic patterns must be segmented and mapped onto internally-stored phonetic and syllabic representations

(*Halle and Stevens, 1962*; *Marslen-Wilson and Welsh, 1978*; *Martin, 2016*). Phonemes must be combined and mapped onto words, which in turn form abstract linguistic structures such as phrases (e.g., *Martin, 2020*; *Pinker and Jackendoff, 2005*).In proficient speakers of a language, this process seems to happen so naturally that one might almost forget the complex parallel and hierarchical processing which occurs during natural speech and language comprehension.

It has been shown that it is essential to track the temporal dynamics of the speech signal in order to understand its meaning (e.g., *Giraud and Poeppel, 2012*; *Peelle and Davis, 2012*). In natural speech, syllables follow each other in the theta range (3–8 Hz; *Rosen, 1992*; *Ding et al., 2017*; *Pellegrino et al., 2011*), while higher-level linguistic features such as words and phrases tend to occur at lower rates (0.5–3 Hz; *Rosen, 1992*; *Kaufeld et al., 2020*; *Keitel et al., 2018*). Tracking of syllabic features is stronger when one understands a language (*Luo and Poeppel, 2007*; *Zoefel et al., 2018a*; *Doelling et al., 2014*) and tracking of phrasal rates is more prominent when the signal contains phrasal information (*Kaufeld et al., 2020*; *Keitel et al., 2018*; *Ding et al., 2016*; e.g., word lists versus sentences). Importantly, phrasal tracking even occurs when there are no distinct acoustic modulations at the phrasal rate (*Kaufeld et al., 2020*; *Keitel et al., 2018*; *Ding et al., 2016*). These results seem to suggest that tracking of relevant temporal timescales is critical for speech understanding.

An observation one could make regarding these findings is that tracking occurs only at the rates that are meaningful and thereby behaviourally relevant (*Kaufeld et al., 2020*; *Ding et al., 2016*). For example, in word lists, the word rate is the slowest rate that is meaningful during natural listening. Modulations at slower phrasal rates might not be tracked as they do not contain behaviourally relevant information. In contrast, in sentences, phrasal rates contain linguistic information and therefore these slower rates are also tracked. Thus, when listening to speech one automatically tries to extract the meaning, which requires extracting information at the highest linguistic level (*Halle and Stevens, 1962*; *Martin, 2016*). However, it remains unclear if tracking at these slower rates is a unique feature of language processing, or rather is dependent on attention to relevant temporal timescales.

As understanding language requires a multitude of processes, it is difficult to figure out what participants actually are doing when listening to natural speech. Moreover, designing a task in an experimental setting that does justice to this multitude of processing is difficult. This is probably why tasks in language studies vary vastly. Tasks include passively listening (e.g., *Kaufeld et al., 2020*), asking comprehension questions (e.g., *Keitel et al., 2018*), rating intelligibility (e.g., *Luo and Poeppel, 2007*; *Doelling et al., 2014*), working memory tasks (e.g., *Kayser et al., 2015*), or even syllable counting (e.g., *Ding et al., 2016*). It is unclear whether outcomes are dependent on the specifics of the task. There has so far not been a study that investigates if task instructions focusing on extracting information at different temporal rates or timescales have an influence on the tracking that occurs on these timescales. It is therefore not clear whether tracking phrasal timescales is unique for language stimuli which contain phrasal structures, or could also occur for other acoustic materials where participants are instructed to pay attention to information happening at these temporal rates or timescales.

To answer this question, we designed an experiment in which participants were instructed to pay attention to different temporal modulation rates while listening to the same stimuli. We presented participants with naturally spoken sentences and word lists and asked them to either passively listen, or perform a task on the temporal scales corresponding to syllables, words, or phrases. We recorded brain activity using magnetoencephalography (MEG) while participants performed these tasks and investigated tracking as well as power and connectivity at three nodes that are part of the language network: the superior temporal gyrus (STG), the middle temporal gyrus (MTG), and the inferior frontal gyrus (IFG). We hypothesized that if tracking is purely based on behavioural relevance, it should mostly depend on the task instructions, rather than the nature of the stimuli. In contrast, if there is something automatic and specific about language information, tracking should depend on the level of linguistic information available to the brain.

## Results

### Behaviour

Overall task performance was above chance and participants complied with task instructions (*Figure 1*; see *Figure 1—figure supplement 1* for individual data). We found a significant interaction between condition and task ($F_{(2,72.0)} = 11.51$, $p < 0.001$) as well as a main effect of task ($F_{(2,19.7)} = 44.19$,

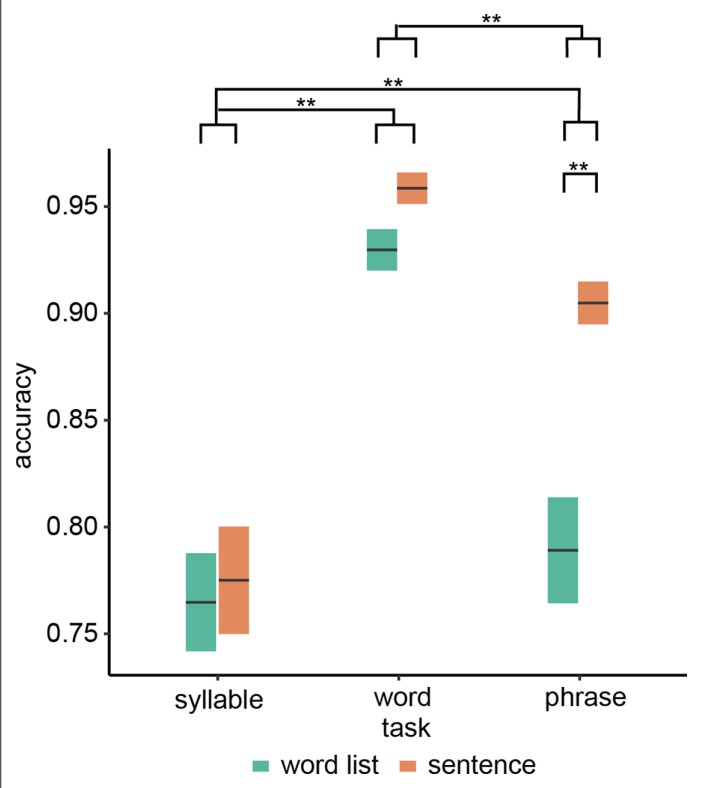

**Figure 1.** Behavioural results. Accuracy for the three different tasks. Double asterisks indicate significance at the 0.01 level using a paired samples t-test (n=19). Box edges indicate the standard error of the mean.

The online version of this article includes the following figure supplement(s) for figure 1:

**Figure supplement 1.** Behavioural results with individual data.

p < 0.001) and condition (*F*(2,72.0) = 29.0, p < 0.001). We found that only for the word-combination (phrasal-level) task, the sentence condition had a significantly higher accuracy than the word list condition (*t*(54.0) = 6.97, p < 0.001). For the other two tasks, no significant condition effect was found (syllable: *t*(54.0) = 0.62, p = 1.000; word list: *t*(54.0) = 1.74, p = 0.176). Investigating the main effect of task indicated a difference between all tasks (phrase–syllable: *t*(18.0) = 3.71, p = 0.003; phrase–word: *t*(22.4) = −6.34, p < 0.001; syllable–word: *t*(19.2) = −8.67, p < 0.001).

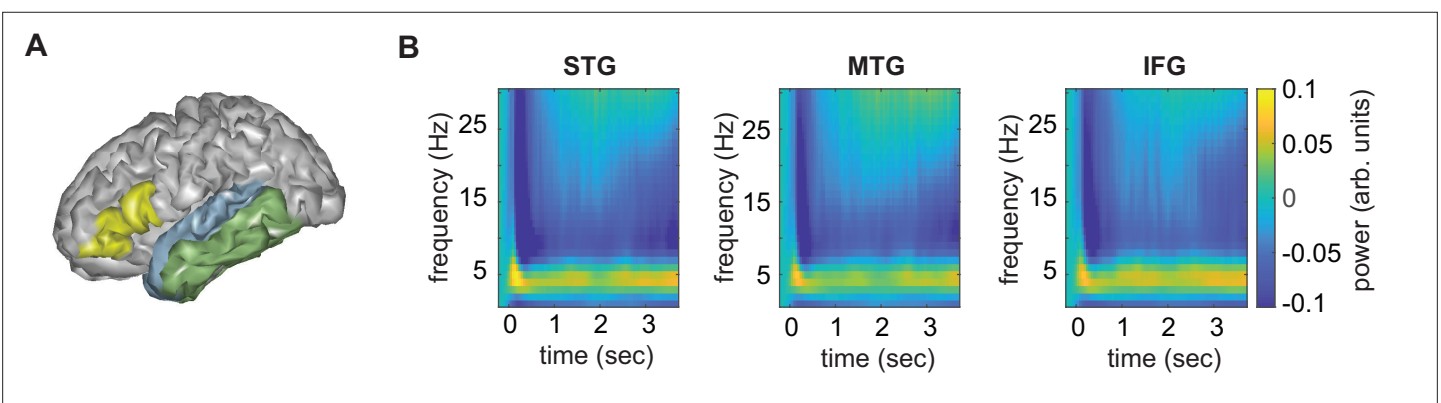

**Figure 2.** Anatomical regions of interests (ROIs). (**A**) ROIs displayed on one exemplar participant surface. (**B**) Time–frequency response at each ROI. STG = superior temporal gyrus, MTG = medial temporal gyrus, IFG = inferior frontal gyrus.

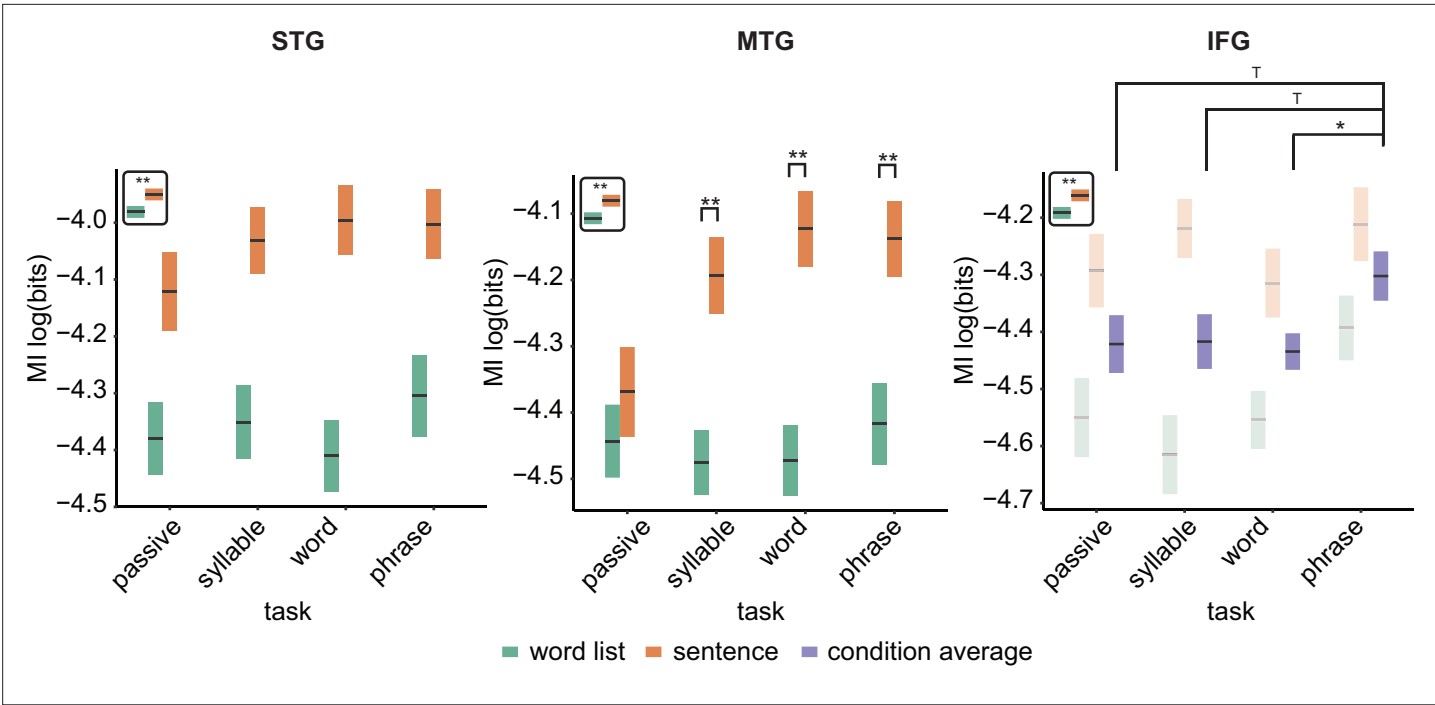

**Figure 3.** Mutual information (MI) analysis at the phrasal band (0.8–1.1 Hz) for the three different regions of interests (ROIs). Single and double asterisks indicate significance at the 0.05 and 0.01 level using a paired samples t-test (n=19). T indicates trend level significance (p < 0.1). Inset at the top left of the graph indicates whether a main effect of condition was present (with higher MI for sentences versus word lists; this inset does not reflect real data). Averages of conditions are only shown if there was a main task effect without an interaction. Box edges indicate the standard error of the mean.

The online version of this article includes the following figure supplement(s) for figure 3:

**Figure supplement 1.** Mutual information (MI) analysis at the phrasal band (0.8–1.1 Hz) for the three different regions of interests (ROIs) with individual data.

**Figure supplement 2.** Mutual information (MI) analysis at the syllable (3.5–5.0 Hz) and word rate (1.9–2.8 Hz) for the three different regions of interests (ROIs).

## Mutual information

The overall time–frequency response in the three different regions of interest (ROI) using the top-20 PCA components was as expected, with an initial evoked response followed by a more sustained response to the ongoing speech (*Figure 2*). From these regions-of-interest, we extracted mutual information (MI) in three different frequency bands (phrasal, word, and syllable). Here, we focus on the phrasal band as this is the band that differentiates word lists from sentences and showed the strongest modulation for this contrast in our previous study (*Kaufeld et al., 2020*). MI results for all other bands are reported in the supplementary materials.

For the phrasal timescale in STG, we found significantly higher MI in the sentence compared to the word list condition ($F(3,126) = 67.39$, $p < 0.001$; *Figure 3*; see *Figure 3—figure supplement 1* for individual data). No other effects were significant ($p > 0.1$). This finding paralleled the effect found in *Kaufeld et al., 2020*. For the MTG, we saw a different picture: Besides the main effect of condition ($F(3,126) = 50.24$, $p < 0.001$), an interaction between task and condition was found ($F(3,126) = 2.948$, $p = 0.035$). We next investigated the effect of condition per task and found for all tasks except the passive task a significant effect of condition, with stronger MI for the sentence condition (passive: $t(126) = 1.07$, $p = 0.865$; syllable: $t(126) = 4.06$, $p = 0.003$; word: $t(126) = 5.033$, $p < 0.001$; phrase: $t(126) = 4.015$, $p = 0.003$). For the IFG, we found a main effect of condition ($F(3,108) = 21.89$, $p < 0.001$) as well as a main effect of task ($F(3,108) = 2.74$, $p = 0.047$). The interaction was not significant ($F(3,108) = 1.49$, $p = 0.220$). Comparing the phrasal task with the other tasks indicated higher MI for the phrasal compared to the word task ($t(111) = 2.50$, $p = 0.028$). We also found a trend for the comparison between the phrasal and syllable tasks ($t(111) = 2.17$, $p = 0.064$), as well as the phrasal and passive tasks ($t(111) = 2.25$, $p = 0.052$).

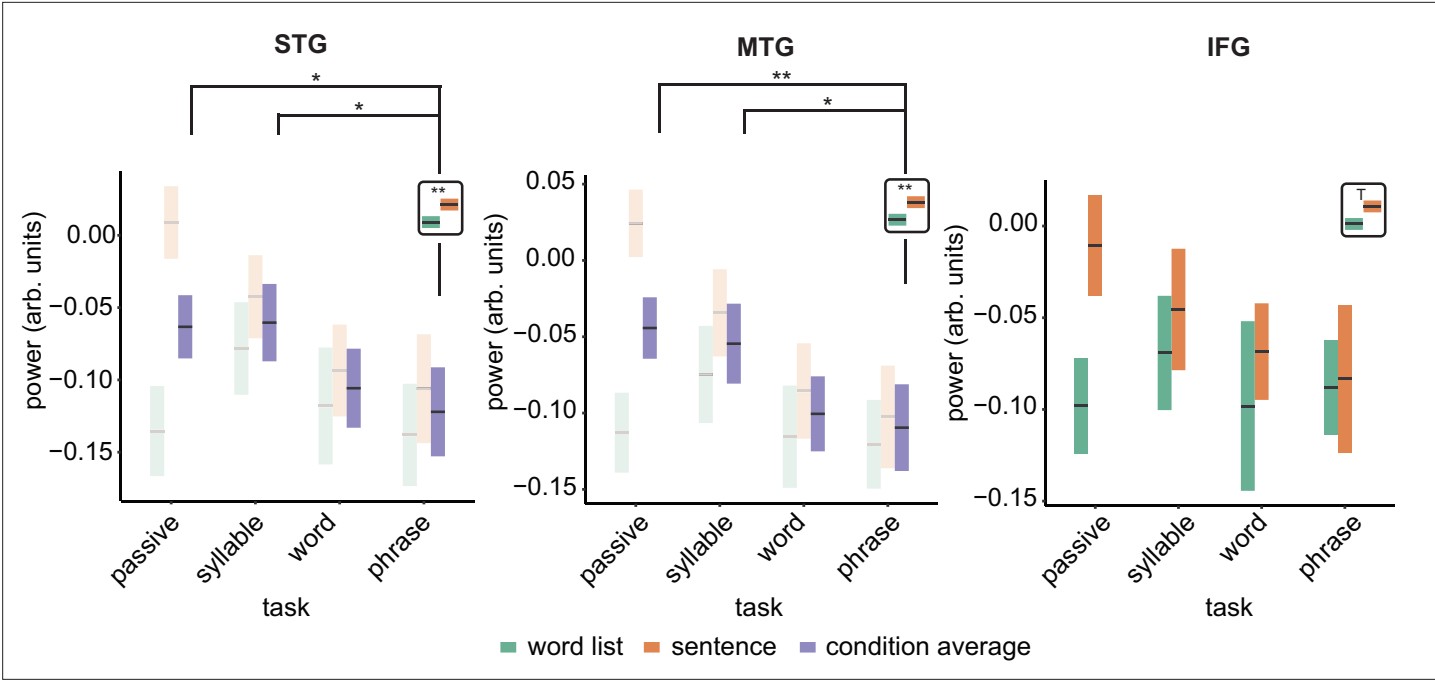

**Figure 4.** Power effects for the different regions of interests (ROIs). Single and double asterisks indicate significance at the 0.05 and 0.01 level using a paired samples t-test (n=19). T indicates trend significance (p < 0.1) Inset at the right top of the graph indicates whether a main effect of condition was present (with higher activity for sentences versus word lists; this inset does not reflect real data). Averages of conditions are only shown if there was a main task effect. Box edges indicate the standard error of the mean.

The online version of this article includes the following figure supplement(s) for figure 4:

**Figure supplement 1.** Power effects for the different regions of interests (ROIs) with individual data.

**Figure supplement 2.** Power effects for the different regions of interests (ROIs) and different bands.

For the word and syllable frequency bands no interactions were found (all p > 0.1; *Figure 3—figure supplement 2*). For all six models, there was a significant effect of condition, with stronger MI for word lists compared to sentences (all p < 0.001). The main effect of task was not significant in any of the models (p > 0.1; for the MTG syllable level there was a trend: $F(3,126) = 2.40$, p = 0.071).

When running the power control analysis, we did not find that significant effects in power differences (also see next section for power in generic bands; mostly due to main effects of condition) influenced our tracking results for any of the bands investigated.

## Power

We repeated the linear mixed modelling using power instead of MI to investigate if power changes paralleled the MI effects. For the delta band, we found for the STG a main effect of condition ($F(1,18) = 6.11$, p = 0.024; *Figure 4*. See *Figure 4—figure supplement 1* for individual data) and task ($F(3,108) = 3.069$, p = 0.031). For the interaction we found a trend ($F(3,108) = 2.620$, p = 0.054). Overall sentences had stronger delta power than word lists. We found lower power for the phrase compared to the passive task ($t(111) = 2.31$, p = 0.045) and lower power for the phrase compared to the syllable task ($t(111) = 2.43$, p = 0.034). There was no significant difference between the phrase and word task ($t(111) = 0.642$, p = 1.00).

The MTG delta power effect overall paralleled the STG effects with a significant condition ($F(1,124.94) = 12.339$, p < 0.001) and task effect ($F(3,124.94) = 4.326$, p = 0.006). The interaction was trend significant ($F(3,124.94) = 2.58$, p = 0.056). Pairwise comparisons of the task effect showed significantly stronger power for the phrase compared to the passive task ($t(128) = 2.98$, p = 0.007) and lower power for the phrase compared to the syllable task ($t(128) = 3.10$, p = 0.024). The passive–word comparison was not significant ($t(128) = 2.577$, p = 0.109). Finally, for the IFG we only found a trend effect for condition ($F(1,123.27)=4.15$, p = 0.057), with stronger delta power in the sentence condition.

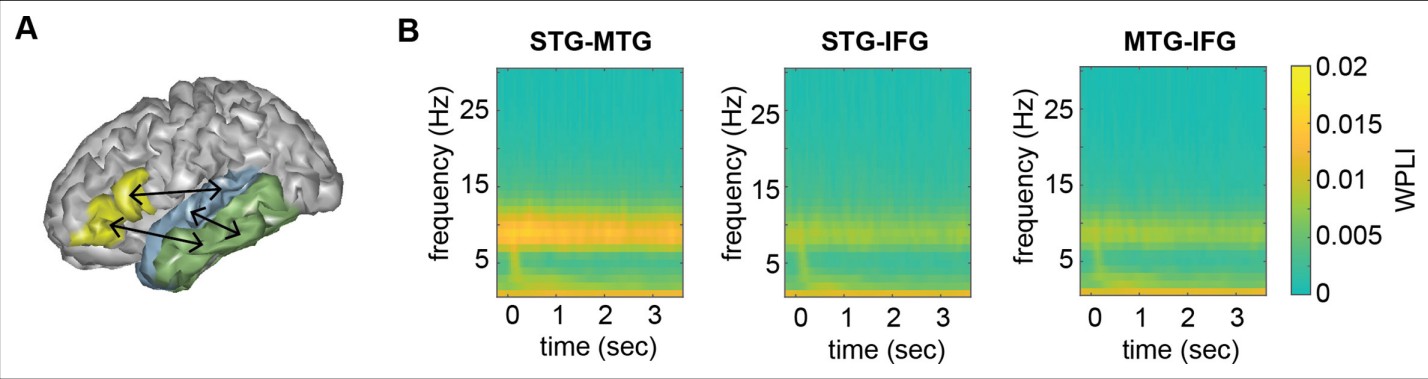

**Figure 5.** Connectivity pattern between anatomical regions of interests (ROIs). (**A**) ROI connections displayed on one exemplar participant surface. (**B**) Time–frequency weighted phase-lagged index (WPLI) response at each ROI.

The results for all other bands can be found in the supplementary materials (*Figure 4—figure supplement 2*). In summary, no interaction effects were found for any of the models (all p > 0.1). In all bands, power was generally higher for sentences than for word lists. Any task effect generally showed stronger power for the lower hierarchical level (e.g. generally higher power for passive versus word-combination tasks).

## Connectivity

Overall connectivity patterns showed the strongest connectivity in the delta and alpha frequency band (*Figure 5*). In the delta band, we found a main effect of task for the STG–IFG connectivity ($F$(3,122.06) = 4.1078, p = 0.008; *Figure 6*; see *Figure 6—figure supplement 1* for individual data). Follow-up analysis showed a significant difference between the phrasal and passive tasks with higher connectivity

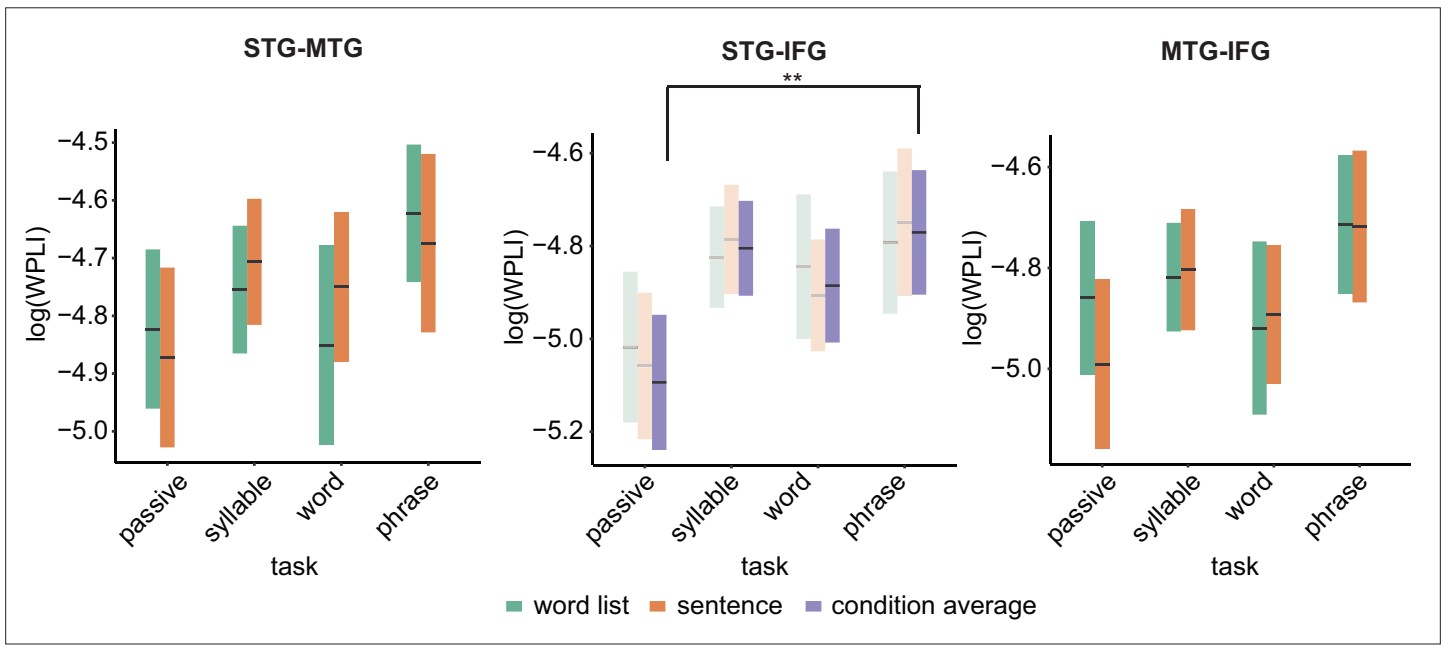

**Figure 6.** Weighted phase lag index (WPLI) effects for the different regions of interests (ROIs). Double asterisks indicate significance at the 0.01 level using a paired samples t-test (n=19) after correcting for power differences between the two conditions (we plot the original data, not corrected for power, as we can only perform pairwise power and consequently data will be different for each control). Averages of conditions are only shown if there was a main task effect. Box edges indicate the standard error of the mean.

The online version of this article includes the following figure supplement(s) for figure 6:

**Figure supplement 1.** Weighted phase lag index (WPLI) effects for the different regions of interests (ROIs) with individual data.

**Figure supplement 2.** Weighted phase lag index (WPLI) effects for the different regions of interests (ROIs) and different bands.

in the phrasal compared to the passive task ($t(125)$ = 3.254, p = 0.003). The other comparisons with the phrasal task were not significant. The effect of task remained significant even when correcting for power differences between the passive and phrasal tasks ($F(1,53.02)$ = 12.39, p < 0.001; note the change in degrees of freedom as only the passive and phrasal tasks were included in this mixed model as any power correction is done on pairs). Initially, we also found main effects of condition for the delta and beta bands for the MTG–IFG connectivity (stronger connectivity for the sentence compared to the word list condition), however after controlling for power, these effects did not remain significant (*Figure 6—figure supplement 2*).

## MEG–behavioural performance relation

We found for the MI analysis a significant effect of accuracy only in the MTG. Here, we found a three-way interaction between accuracy × task × condition ($F(2,91.9)$ = 3.459, p = 0.036). Splitting up for the three different tasks we found only an uncorrected significant effect for the condition × accuracy interaction for the phrasal task ($F(1,24.8)$ = 5.296, p = 0.03) and not for the other two tasks

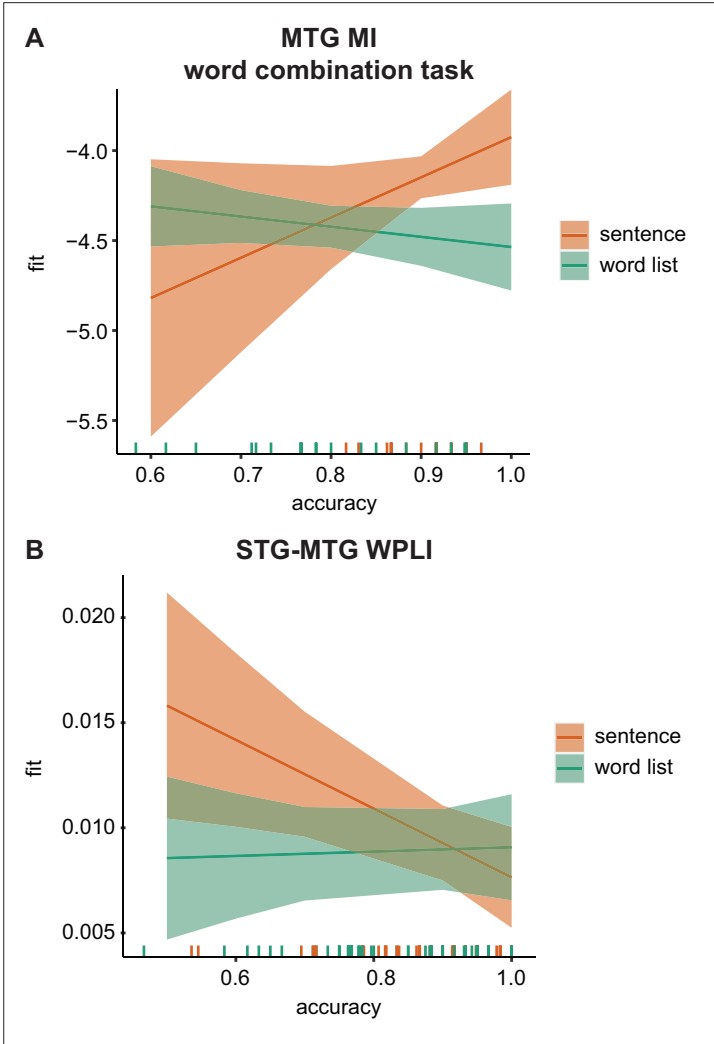

**Figure 7.** MEG–behavioural performance relation. (**A**) Predicted values for the phrasal band MI in the middle temporal gyrus (MTG) for the word-combination task separately for the two conditions. (**B**) Predicted values for the delta-band weighted phase lag index (WPLI) in the superior temporal gyrus (STG)–MTG connection separately for the two conditions. Error bars indicate the 95% confidence interval of the fit. Coloured lines at the bottom indicate individual datapoints.

The online version of this article includes the following figure supplement(s) for figure 7:

**Figure supplement 1.** Age effects on power estimates.

(p > 0.1). In the phrasal task, we found that when accuracy was high, there was a stronger difference between the sentence and the word list condition compared to when accuracy was low, with stronger accuracy for the sentence condition (*Figure 7A*).

No relation between accuracy and power was found. For the connectivity analysis, we found a significant condition × accuracy interaction for the STG–MTG connection ($F(1,80.23)$ = 5.19, p = 0.025; *Figure 7B*). Independent of task, when accuracy was low the difference between sentence and word lists was stronger with higher weighted phase lag index (WPLI) fits for the sentence condition. After correcting for accuracy there was also a significant task × condition interaction ($F(2,80.01)$ = 3.348, p = 0.040) and a main effect of condition ($F(1,80.361)$ = 5.809, p = 0.018). While overall there was a stronger WPLI for the sentence compared to the word list condition, the interaction seemed to indicate that this was especially the case during the word task (p = 0.005), but not for the other tasks (p > 0.1).

### Age control

Adding age to the analysis did not change any of the original findings (all original effects were still significant). We did however find for the power analysis age-specific interactions with condition and task. Specifically, for both the STG and the MTG we found an interaction between age and condition ($F(1,28.87)$ = 6.156, p = 0.0192 and $F(1,31)$ = 10.31, p = 0.003). In both ROIs, there was a stronger difference between sentences and word lists (higher delta power for sentences) for the younger compared to the older participants (*Figure 7—figure supplement 1*). In the MTG, there was also an interaction between task and age ($F(1,31)$ = 5.020, p = 0.006). Here, in a follow-up we found that only in the word task there was a correlation between age and power (p = 0.023 uncorrected), but not for the other tasks (p > 0.1).

### Number of component control

Overall, the amount of PCA components did not influence any of the qualitative differences in the condition. It did seem however that 10 PCA components were not sufficient to show all original effects with the same power. Specifically, the IFG task and MTG task × condition effect were only trend significant for 10 components (p = 0.06 and p = 0.1, respectively). The other effects did remain significant with 10 components. Using 30 components made some of our effects stronger than with 20 components. Here, the IFG task and MTG task × condition effects had p values of 0.034 and 0.006, respectively. We conclude that the amount of PCAs components did not qualitative change any of our reported effects.

## Discussion

In the current study, we investigated the effects of 'additional' tasks on the neural tracking of sentences and word lists at temporal modulations that matched phrasal rates. Different nodes of the language network showed different tracking patterns. In STG, we found stronger tracking of phrase-timed dynamics in sentences compared to word lists, independent of task. However, in MTG we found this sentence-improved tracking only for active tasks. In IFG, we also found an overall increase of tracking for sentences compared to word lists. Additionally, stronger phrasal tracking was found for the phrasal-level word-combination task compared to the other tasks (independent of stimulus type; note that for the syllable and passive comparison we found a trend), which was paralleled with increased IFG–STG connectivity in the delta band for the word-combination task. Behavioural performance seemed to relate to MI tracking in the MTG and STG–MTG connections. This suggests that tracking at phrasal timescales depends both on the linguistic information present in the signal, and on the specific task that is performed.

The findings reported in this study are in line with previous results, with overall stronger tracking of low-frequency information in the sentences compared to the word list condition (*Kaufeld et al., 2020*). Crucially, for the stimuli used in our study it has been shown that the condition effects are not due to acoustic differences in the stimuli and also do not occur for reversed speech (*Kaufeld et al., 2020*). It is therefore most likely that our results reflect an automatic inference-based extraction of relevant phrase-level information in sentences, indicating automatic processing in participants as they understand the meaning of the speech they hear using stored, structural linguistic knowledge (*Martin,*

*2020*; *Ding et al., 2016*; *Har-Shai Yahav and Zion Golumbic, 2021*). Overall, it did not seem that making participants pay attention to the temporal dynamics at the same hierarchical level through an additional task – instructing them to remember word combinations at the phrasal rate during word list presentation – could counter this main effect of condition.

Even though there was an overall main effect of condition, task did influence neural responses. Interestingly, the task effects differed for the three ROIs. In the STG, we found no task effects, while in the MTG we found an interaction between task and condition. In the MTG increased phrasal-level tracking for sentences only occurred when participants were specifically instructed to perform an active task on the materials. It therefore seems that in MTG all levels of linguistic information are used to do an active language operation on the stimuli. Importantly, the tracking at the phrasal rate in MTG seemed relevant for behavioural performance when attending to phrasal timescales (*Figure 7A*). This is in line with previous theoretical and empirical research suggesting a strong top-down modulatory response of speech processing in which predictions flow from the highest hierarchical levels (e.g., syntax) down to lower levels (e.g., phonemes) to aid language understanding (*Martin, 2016*; *Hagoort, 2017*; *Federmeier, 2007*). As in the word list condition no linguistic information is present at the phrasal rate, this information cannot be used to provide useful feedback for processing lower-level linguistic information. Instead, it could have been expected that the same type of increased tracking should have happened at the word rate rather than the phrasal rate for word lists (i.e., stronger word-rate tracking for word lists for the active tasks versus passive task). This effect was not found; this could either be attributed to different computational operations occurring at different hierarchical levels or to signal-to-noise/signal detection issues.

We found that across participants both the MI and the connectivity in temporal cortex influenced behavioural performance. Specifically, MTG–STG connections were, independent of task, related to accuracy. There was higher connectivity between MTG and STG for sentences compared to word lists at low accuracies. At high accuracies, we found that stronger MTG tracking at phrasal rates (measured with MI) for sentences compared to word lists during the word-combination task. These results suggest that indeed tracking of phrasal structure in MTG is relevant to understand sentences compared to word lists. This was reflected in a general increase in delta connectivity differences when the task was difficult (*Figure 7B*). Participants might compensate for the difficulty using phrasal structure present in the sentence condition. When phrasal structure in sentences are accurately tracked (as measured with MI) performance is better when these rates are relevant (*Figure 7A*). These results point to a role for phrasal tracking for accurately understanding the higher-order linguistic structure in sentences, though more research is needed to verify this. It is evident that the connectivity and tracking correlations to behaviour do not explain all variation in the behavioural performance (compare *Figure 1* with *Figure 3*). Plainly, temporal tracking does not explain everything in language processing. Besides tracking there are many other components important for our designated tasks, such as memory load and semantic context which are not captured by our current analyses.

It is interesting that MTG, but not STG, showed an interaction effect. Both MTG and STG are strong hubs for language processing and have been involved in many studies which contrasted pseudowords and words (*Hickok and Poeppel, 2007*; *Turken and Dronkers, 2011*; *Vouloumanos et al., 2001*). It is likely that STG does the more lower-level processing of the two regions, as it is earlier in the cortical hierarchy, thereby being more involved in initial segmentation and initial phonetic abstraction rather than a lexical interface (*Hickok and Poeppel, 2007*). This could also explain why STG does not show task-specific tracking effects; STG could be earlier in a workload bottleneck, receiving feedback independent of task, while MTG feedback is recruited only when active linguistic operations are required. Alternatively, it is possible that either small differences in the acoustics are detected by STG (even though this effect was not previously found with the same stimuli, *Kaufeld et al., 2020*), or that our blocked designed put participants in a sentence or word list 'mode' which could have influenced the state of these early hierarchical regions.

The IFG was the only region that showed an increase in phrasal-rate tracking specifically for the word-combination task. Note, however, that this was a weak effect, as the comparison between the phrase task and the syllable and passive tasks only reached a trend towards significance. Nonetheless, this effect is interesting for understanding the role of IFG in language. Traditionally, IFG has been viewed as a hub for articulatory processing (*Hickok and Poeppel, 2007*), but its role during speech comprehension, specifically in syntactic processing, has also been acknowledged (*Friederici, 2011*;

*Hagoort, 2017*; *Nelson et al., 2017*; *Dehaene et al., 2015*; *Zaccarella et al., 2017*). Integrating information across time and relative timing is essential for syntactic processing (*Martin, 2020*; *Dehaene et al., 2015*; *Martin and Doumas, 2019*), and IFG feedback has been shown to occur in temporal dynamics at lower (delta) rates during sentence processing (*Park et al., 2015*; *Keitel and Gross, 2016*). However, it has also been shown that syntactic-independent verbal working memory chunking tasks recruit the IFG (*Dehaene et al., 2015*; *Osaka et al., 2004*; *Fegen et al., 2015*; *Koelsch et al., 2009*). This is in line with our findings that show that IFG is involved when we need to integrate across temporal domains either in a language-specific domain (sentences versus word lists) or for language-unspecific tasks (word-combination versus other tasks). We also show increased delta connectivity with STG for the only temporal-integration tasks in our study (i.e., the word-combination task), independent of the linguistic features in the signal. Our results therefore support a role of the IFG as a combinatorial hub integrating information across time (*Gelfand and Bookheimer, 2003*; *Schapiro et al., 2013*; *Skipper, 2015*).

In the current study, we investigated power as a neural readout during language comprehension from speech. This was both to ensure that any tracking effects we found were not due to overall signal-to-noise (SNR) differences, as well as to investigate task-and-condition dependent computations. SNR is better for conditions with higher power, which therefore leads to more reliable phase estimations, critical for computing MI as well as connectivity (*Zar, 1998*). We will therefore discuss the power differences as well as their consequences for the interpretation of the MI and connectivity results. Generally, it seemed that there was stronger power in the sentence condition compared to the word list condition in the delta band. However, the pattern was very different than the MI pattern. For the power, the word list-sentence difference was the biggest in the passive condition. In contrast, for the MI there was either no task difference (in STG) or even a stronger effect for the active tasks (in MTG; note that the power interaction was trend significant STG and MTG). We therefore think it unlikely that our MI effects were purely driven by SNR differences, and our power control analysis is consistent with this interpretation. Instead, power seems to reflect a different computation than the tracking, where more complex tasks generally lead to lower power across almost all tested frequency bands. As most of our frequency bands are on the low side of the spectrum (up to beta), it is expected that more complex tasks reduce the low-frequency power (*Jensen and Mazaheri, 2010*; *Klimesch, 1999*). It is interesting to observe that this did not reduce the connectivity for the delta band between IFG and STG, but rather increased it. It has been suggested that low power can potentially increase the available computational space, as it increases the entropy in the signal (*Hanslmayr et al., 2012*; *ten Oever and Sack, 2015*). Note that even though we found increased connectivity, we did not see a clear power peak in the delta band. This suggests that we might not be looking at an endogenous oscillator, but rather at connections operation at that temporal scale (potentially being non-oscillatory in nature). Finally, in the power comparisons for the theta, alpha, and beta bands we found stronger power for the sentence compared to the word list condition, which could reflect that listening to a natural sentence is generally less effortful than listening to a word list.

In the current manuscript, we describe tracking of ongoing temporal dynamics. However, the neural origin of this tracking is unknown. While we can be sure that modulations in the phrasal-rate follow changes in the phrasal rate of the acoustic input, it is unclear what the mechanism behind this modulation is. It is possible that there is stronger alignment of neural oscillations with the acoustic input at the phrasal rate (*Lakatos et al., 2008*; *Obleser and Kayser, 2019*; *Rimmele et al., 2021*). However, it could as well be that there is a phrasal timescale or slower operation happening while processing the incoming input (which de facto is at the same timescale as the phrasal structure inferred from the input). This operation, in response to stimulus input, could just as well induce the patterns we observe (*Meyer et al., 2019*; *Zoefel et al., 2018b*). Finally, it is possible that there are specific responses as a consequence of the syntactic structure, task, or statistical regularities occurring as specific events at phrasal timescales (*Obleser and Kayser, 2019*; *Ten Oever and Martin, 2021*; *Frank and Yang, 2018*).

It is difficult to decide on the most natural task in an experimental setting, that best reflects how we use language in a natural setting. This is probably why such a vast number of different tasks have been used in the literature. Our study (and many before us) indicates that during passive listening, we naturally attend to all levels of linguistic hierarchy. This is consistent with the widely accepted notion that the meaning of a natural sentence requires composing words in a grammatical structure.

**Table 1.** Stimuli and task examples.

| | Sentence | | Word list | |
|---|---|---|---|---|
| Sentence | [bange helden] [plukken bloemen] en de [bruine vogels] [halen takken] [*timid heroes*] [*pluck flowers*] *and the* [*brown birds*] [*gather branches*] | | | |
| Word list | [helden bloemen] [vogels takken] de en [plukken halen] [bange bruine] [*heroes flowers*] [*birds branches*] *and the* [*pluck gather*] [*timid brown*] | | | |
| | **Sentence** | | **Word list** | |
| | Correct | Incorrect | Correct | Incorrect |
| Syllable | /bɑ/ | /lɑ/ | /bɑ/ | /lɑ/ |
| Word | bloemen [*flowers*] | vaders [*fathers*] | bloemen [*flowers*] | vaders [*fathers*] |
| Word combination | bange helden [*timid heroes*] | halen bloemen [*gather flowers*] | helden bloemen [*heroes flowers*] | vogels bloemen [*birds flowers*] |

For each condition (sentence and word list) one example stimulus (top) and corresponding tasks are shown (bottom).

For most research questions in language, it therefore is sensible to use a task that mimics this automatic natural understanding of a sentence. Here, we show that automatic understanding of linguistic information, and all the processing that this entails, cannot be countered to substantially change the consequences for neural readout, even when explicitly instructing participants to pay attention to particular timescales.

## Materials and methods
### Participants
In total, 20 Dutch native speakers (16 females; age range: 18–59; mean age = 39.5) participated in the study. All were right handed, reported normal hearing, had normal or corrected-to-normal vision, and did not have any history of dyslexia or other language-related disorders. Participants performed a screening for their eligibility in the MEG and MRI and gave written informed consent. The study was approved by the Ethical Commission for human research Arnhem/Nijmegen (project number CMO2014/288). Participants were reimbursed for their participation. One participant was excluded from the analysis as they did not finish the full session.

### Materials and design
Materials were identical to the stimuli used in *Kaufeld et al., 2020*. They consisted of naturally spoken sentences or word lists which consisted of 10 words (see *Table 1* for examples). The sentences contained two coordinate clauses with the following structure: [Adj N V N Conj Det Adj N V N]. All words were disyllabic except for the words 'de' (*the*) and 'en' (*and*). Word lists were word-scrambled versions of the original sentences which always followed the structure [V V Adj Adj Det Conj N N N N] or [N N N N Det Conj V V Adj Adj] to ensure that they were grammatically incorrect. In total, 60 sentences were used. All sentences were presented at a comfortable sound level.

Participants were asked to perform four different tasks on these stimuli: a passive task, a syllable task, a word task, and a word-combination task. For the passive task, participants did not need to perform any task other than comprehension – they only needed to press a button to go to the next trial. For the syllable task, participants heard after every sentence two part-of-speech sounds, each consisting of one syllable. The sound fragments were a randomly determined syllable from the previously presented sentence and a random syllable from all other sentences. Participants' task was to indicate via a button press which of the two sound fragments was part of the previous sentence. For the word task, two words were displayed on the screen after each trial (a random word from the just presented sentence and one random word from all other sentences excluding 'de' and 'en'), and participants needed to indicate which of the two words was part of the sentence before. For the word-combination task, participants were presented with two word pairs on the screen. Each of the four words was part of the just presented sentence, but only one of the pairs was in the correct order. Participants needed to indicate which of the two pairs was presented in the sentence before.

Presented options for the sentence condition were always a grammatically and semantically plausible combination of words. See *Table 1* for an example of the tasks for each condition (sentences and word lists). The three active tasks required participants to focus on the syllabic (syllable task), word (word task), or phrasal (word combination or also called phrasal task) timescales. Note that different trials within a task were not matched for task difficulty. For example, in the syllable task syllables that make a word are much easier to recognize than syllables that do not make a word. Additionally, trials pertaining to the beginning of the sentence are more difficult than ones related to the end of the sentence due to recency effects.

## Procedure

At the beginning of each trial, participants were instructed to look at a fixation cross presented at the middle of the screen on a grey background. Audio recordings were presented after a random interval between 1.5 and 3 s; 1 s after the end of the audio, the task was presented. For the word and word-combination task, this was the presentation of visual stimuli. For the syllable task, this entailed presenting the sound fragments one after each other (with a delay of 0.5 s in between). For the passive task, this was the instruction to press a button to continue. In total, there were eight blocks (two conditions × four tasks) each lasting about 8 min. The order of the blocks was pseudo-randomized by independently randomizing the order of the tasks and the conditions. For a single participant, we then always presented the same task twice in a row to avoid task-switching costs. As a consequence, condition was always alternated (a possible order of blocks would be: passive-sentence, passive-word list, word-sentence, word-word list, syllable-sentence, syllable-word list, word-combination-sentence, word-combination-word list). Across participants the starting condition was counterbalanced. After the main experiment, an auditory localizer was collected which consisted of listening to 200 ms sinewave and broadband sounds (centred at 0.5, 1, and 2 kHz; for the broadband at a 10% frequency band) at approximately equal loudness. Each sound had a 50 ms linear on and off ramp and was presented for 30 times (with random inter-stimulus interval between 1 and 2 s).

At arrival, participants filled out a screening. Electrodes to monitor eye movements and heart beat were placed (left mastoid was used as ground electrode) at an impedance below 15 kΩ. Participants wore metal free clothes and fitted earmolds on which two of the three head localizers were placed (together with a final head localizer placed at the nasion). They then performed the experiment in the MEG. MEG was recorded using a 275-channel axial gradiometer CTF MEG system at a sampling rate of 1.2 kHz. After every block participants had a break, during which head position was corrected (*Stolk et al., 2013*). After the session, the headshape was collected using Polhemus digitizer (using as fiducials the nasion and the entrance of the ear canals as positioned with the earmolds). For each participant, an MRI was collected with a 3T Siemens Skyra system using the MPRAGE sequence (1 mm isotropic). Also for the MRI acquisition participants wore the earmolds with vitamin pills to optimize the alignment.

## Behavioural analysis

We performed a linear mixed model analysis with fixed factors task (syllable, word, and word combination) and condition (sentence and word list) as implemented by lmer in R4.1.0. The dependent variable was accuracy. First, any outliers were removed (values more extreme than median ± 2.5 IQR). Then, we investigated what the best random model was, including a random intercept or a random slope for one or two of the factors. The models with varying random factors were compared with each other using an analysis of variance. With no significant difference, the model with the lowest number of factors was included (with minimally a random intercept). Finally, lsmeans was used for follow-up tests using the kenward-roger method to calculate the degrees of freedom from the linear mixed model. For significant interactions, we investigated the effect of condition per task. For main effects, we investigated pairwise comparisons. We corrected for multiple comparisons using adjusted Bonferroni corrections unless specified otherwise. For all further reported statistical analyses for the MEG data, we followed the same procedure (except that there was one more level of task, i.e. the passive task). To avoid exploding the amount of comparisons, we a priori decided for any task effects in the MEG analysis to only compare the individual tasks with the phrase task.

## MEG pre-processing

First source models from the MRI were made using a surface-based approach in which grid points were defined on the cortical sheet using the automatic segmentation of freesurfer6.0 (*Fischl, 2012*) in combination with pre-processing tools from the HCP workbench1.3.2 (*Glasser et al., 2013*) to down-sample the mesh to 4 k vertices per hemisphere. The MRI was co-registered to the MEG using the previously defined fiducials as well as an automatic alignment of the MRI to the Polhemus headshape using the Fieldtrip20211102 software (*Oostenveld et al., 2011*).

Pre-processing involved epoching the data between −3 and +7.9 s (+3 relative to the longest sentence of 4.9 s) around sentence onset. We applied a dftfilter at 50, 100, and 150 Hz to remove line noise, a Butterworth bandpass filter between 0.6 and 100 Hz, and performed baseline correction (−0.2 to 0 s baseline). Trials with excessive movements or squid jumps were removed via visual inspection (20.1 ± 18.5 trials removed; mean ± standard deviation). Then data were resampled to 300 Hz and we performed ICA decomposition to correct for eye blinks/movement and heart beat artefacts (4.7 ± 0.99 components removed; mean ± standard deviation). Trials with remaining artefacts were removed by visual inspection (11.3 ± 12.4 trials removed; mean ± standard deviation). Then we applied a lcmv filter to transform the data to have single-trial source space representations. A common filter across all trials was calculated using a fixed orientation and a lambda of 5%. We only extracted time courses for our ROI, STG (*Friederici, 2011*; *Park et al., 2015*; *Fegen et al., 2015*; *Koelsch et al., 2009*), medial temporal gyrus (*Pinker and Jackendoff, 2005*; *Peelle and Davis, 2012*; *Luo and Poeppel, 2007*), and inferior frontal cortex (*Ding et al., 2016*; *Kayser et al., 2015*; *Har-Shai Yahav and Zion Golumbic, 2021*); numbers correspond to label-coding from the aparc parcellations implemented in Freesurfer. These time courses were baseline corrected (−0.2 to 0 s). To reduce computational load and to ensure that we used relevant data within the ROI, we extracted the top 20 PCA components per ROI for all following analyses based on a PCA using the time window of interest (0.5–3.7 s; 0.5 to ensure that all initial evoked responses were not included and 3.7 as it corresponds to the shortest trials). All following analyses were done per ROI. With enough statistical power one would add ROI as a separate factor in the analyses, but unfortunately, we did not have enough power to find a potential three-way interaction (ROI × condition × task). We therefore cannot make strong conclusions about one ROI having a stronger effect than another.

## MI analysis

First, we extracted the speech envelopes by following previous procedures (*Kaufeld et al., 2020*; *Keitel et al., 2018*; *Gross et al., 2013*; *Ince et al., 2017*). The acoustic waveforms (third-order Butterworth filter) were filtered in eight frequency bands (100–8000 Hz) equidistant on the cochlear frequency map (*Smith et al., 2002*). The absolute of the Hilbert transform was computed, we low passed the data at 100 Hz (third order Butterworth) and then down-sampled to 300 Hz (matching the MEG sampling rate). Then, we averaged across all bands.

MI was calculated between the filtered speech envelopes (using a third-order Butterworth filter) and the filtered MEG data at three different frequency bands corresponding to information content at different linguistic hierarchical levels: phrase (0.8–1.1 Hz), word (1.9–2.8 Hz), and syllable (3.5–5.0 Hz). The frequency bands were extracted based on the rate of the linguistic information in the speech signal. We hypothesized that tracking of relevant information should happen at those respective bands. While in our stimulus set the boundaries of the linguistic levels did not overlap, in natural speech the brain has an even more difficult task as there is no one-to-one match between band and linguistic unit (*Obleser et al., 2012*). Our main analysis focuses on the phrasal band, as that is where our previous study found the strongest effects (*Kaufeld et al., 2020*), but for completeness we also report on the other bands. MI was estimated after the evoked response (0.5 s) until the end of the stimulus at five different delays (60, 80, 100, 120, and 140 ms) and averaged across delays between the phase estimations of the envelopes and MEG data. A single MI value was generated per condition per ROI by concatenating all trials before calculating the MI (MEG and speech). Statistical analysis was performed per ROI per frequency band.

## Power analysis

Power analysis was performed to compare the MI results with absolute power changes. On the one hand, we did this analysis as MI differences could be a consequence of signal-to-noise differences in

the original data (which would be reflected in power effects). On the other hand, generic delta power has been associated with language processing (*Meyer, 2018*; *Kazanina and Tavano, 2021*). Therefore, we choose to analyse classical frequency bands instead of the stimulus informed ones (as used in the MI analysis) in order to compare these results with other studies. Moreover, as this analysis does not measure tracking to the stimulus (like the MI analysis does) it did not seem appropriate to match the frequency content to the stimulus content. We first extracted the time–frequency representation for all conditions and ROIs separately. To do so, we performed a wavelet analysis with a width of 4, with a frequency of interest between 1 and 30 (step size of 1) and time of interest between −0.2 and 3.7 s (step size of 0.05 s). We extracted the logarithm of the power and baseline corrected the data in the frequency domain using a −0.3 and −0.1 s window. For four different frequency bands (delta: 0.5–3.0 Hz; theta: 3.0–8.0 Hz; alpha: 8.0–15.0 Hz; beta: 15.0–25.0 Hz) we extracted the mean power in the 0.5–3.7 s time window per task, condition, and ROI. Again, our main analysis focuses on the delta band, but we also report on the other bands for completeness. For each ROI, we performed the statistical analysis on power as described in the behavioural analysis.

## Connectivity analysis

For the connectivity analysis, we repeated all pre-processing as in the power analysis, but separately for the left and right hemispheres (as we did not expect connections for PCA across hemispheres), after which we averaged the connectivity measure across hemispheres (using the Fourier spectrum and not the power spectrum). We used the debiased WPLI for our connectivity measure, which ensures that no zero-lag phase differences are included in the estimation (avoiding effects due to volume conduction). All connections between the three ROIs were investigated for the mean WPLI for the four different frequency bands also used in the power analysis in the 0.5–3.7 s time window. Also in this case, the same statistical analysis was applied. Note that not for all frequency bands we found a clear peak in the power signal (only clearly so for the alpha band), this indicates that the connectivity likely does not reflect endogenous oscillatory activity (*Donoghue et al., 2020*), but might still pertain to connected regions operating at those timescales.

## MEG–behavioural performance analysis

To investigate the relation between the MEG measures and the behavioural performance we repeated the analyses (MI, power, and connectivity) but added accuracy as a factor (together with the interactions with the task and condition factor). As there is no accuracy for the passive task, we removed this task from the analysis. We then followed the same analyse steps as before. Since we reduced our degree of freedom, we could however only create random intercept and not random slope models.

## Power control analysis

The reliability of phase estimations is influenced by the signal-to-noise ratio of the signal (*Zar, 1998*). As a consequence, trials with generally high power have more reliable phase estimations compared to low power trials. This could influence any measure relying on this phase estimation, such as MI and connectivity (*Ince et al., 2017*; *Bastos and Schoffelen, 2015*). It is therefore possible that power differences between conditions lead to differences between connectivity or MI. To ensure that our reported effects are not due to signal-to-noise effects, we controlled any significant power difference between conditions for the connectivity and MI analysis. To do this, we iteratively removed the highest and lowest power trials between the mean highest and mean lowest of the two relevant conditions (either collapsing trials across tasks/conditions or using individual conditions; for the MI analysis we used power estimated within its respective frequency band). We repeated this until the original condition with the highest power had lower power than the other condition. Then we repeated the analysis and statistics, investigating if the effect of interest was still significant. The control analysis is reported along the main MI and connectivity sections.

## Other control analysis

We performed two final control analyses. Firstly, we investigated if age had an influence on any of our primary outcome measures. Secondly, we repeated the analyses using either 10 or 30 PCA components instead of the original 20 components. These controls ensure a robustness check of all the reported results. Note that this study was not intended to investigate age related differences.

We therefore only report on interaction effect with our task and condition variables. Any main effect of age is difficult to interpret as it is unclear if the effect pertains to overall age-related differences (or anatomical variation leading to differential MEG responses) or language-related age differences.

## Additional information

### Competing interests

Andrea E Martin: Reviewing editor, eLife. The other authors declare that no competing interests exist.

### Funding

| Funder | Grant reference number | Author |
| --- | --- | --- |
| Max-Planck-Gesellschaft | Lise Meitner Research Group "Language and Computation in Neural Systems" | Andrea E Martin |
| Max-Planck-Gesellschaft | Independent Research Group "Language and Computation in Neural Systems" | Andrea E Martin |
| Nederlandse Organisatie voor Wetenschappelijk Onderzoek | 016.Vidi.188.029 | Andrea E Martin |

The funders had no role in study design, data collection, and interpretation, or the decision to submit the work for publication.

### Author contributions

Sanne ten Oever, Conceptualization, Data curation, Formal analysis, Investigation, Methodology, Software, Supervision, Validation, Visualization, Writing - original draft, Writing – review and editing; Sara Carta, Data curation, Investigation, Methodology, Software, Writing – review and editing; Greta Kaufeld, Conceptualization, Data curation, Methodology, Software, Writing – review and editing; Andrea E Martin, Conceptualization, Formal analysis, Funding acquisition, Project administration, Resources, Software, Supervision, Validation, Visualization, Writing – review and editing

### Author ORCIDs

Sanne ten Oever (ORCID) http://orcid.org/0000-0001-7547-5842
Andrea E Martin (ORCID) http://orcid.org/0000-0002-3395-7234

### Ethics

Participants performed a screening for their eligibility in the MEG and MRI and gave written informed consent. The study was approved by the Ethical Commission for human research Arnhem/Nijmegen (project number CMO2014/288). Participants were reimbursed for their participation.

### Decision letter and Author response

Decision letter https://doi.org/10.7554/eLife.77468.sa1
Author response https://doi.org/10.7554/eLife.77468.sa2

## Additional files

### Supplementary files
• MDAR checklist

### Data availability

Data and analysis code are available at https://data.donders.ru.nl/collections/di/dccn/DSC_3027006.01_220 (doi: https://doi.org/10.34973/vjw9-0572).

The following dataset was generated:

| Author(s) | Year | Dataset title | Dataset URL | Database and Identifier |
|---|---|---|---|---|
| ten Oever S, Carta S, Kaufeld G, Martin AE | 2022 | Task relevant tracking of hierarchical linguistic structure | https://doi.org/10.34973/vjw9-057 | Donders Repository, 10.34973/vjw9-0572 |

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
