## [Editor Report]

This MEG study elegantly assesses human brain responses to spoken language at the syllable, word, and sentence level. Although prior studies have shown significant cortical tracking of the speech signal, the current work uses clever task manipulation to direct attention to different timescales of speech, thus demonstrating tracking mechanisms that are both automatic and task-dependent operate in tandem during spoken language comprehension.

---

## [Decision Letter]

**Decision letter after peer review:**

Thank you for submitting your article "Task-dependent and automatic tracking of hierarchical linguistic structure" for consideration by *eLife*. Your article has been reviewed by 3 peer reviewers, one of whom is a member of our Board of Reviewing Editors, and the evaluation has been overseen by Barbara Shinn-Cunningham as the Senior Editor. The following individual involved in the review of your submission has agreed to reveal their identity: Johanna Rimmele (Reviewer #2).

Essential revisions:

1. The relationship of participants' behavior to the observed MEG responses. One clear concern is the degree to which the accuracy differences between word lists and sentences (Figure 1) explain MI differences.

2. A number of important details regarding analysis choices were missing and should be added, including: justification of the frequency bands used for the MI analysis, task order (were sentences always first, and word lists second), whether ROIs were defined across hemispheres, details on signal filtering, and details regarding the PCA analysis.

3. The use of "hierarchical linguistic structure" should be clarified or tempered in the context of the current results. Although the attention was directed to syllables, words, and phrases through instructions, the significant MEG results were in the phrasal band (Figure 3) and not the others (supplemental figures).

*Reviewer #1 (Recommendations for the authors):*

The participants ranged in age from 18-59, which is a rather broad range. Given age-related changes in both hearing (not assessed) and language processing, it was surprising to not at least see age included in the models (though perhaps there were not enough participants?). Some comment would be useful here.

Figures: The y axis label seems frequently oddly positioned at the top – maybe consider just centering it on the axis?

I liked the inset panels for the main effects, but they are rather small. If you can make them any larger that would improve readability.

In general, I really like showing individual subjects in addition to means. However, it also can make the plots a bit busy. I imagine you've tried other ways of plotting but might be worth exploring ways to highlight the means a bit more (or simplify the main paper figures, and keep these more detailed figures for supplemental)? Totally up to you, but I had a hard time seeing the trends as the plots currently stand.

On my copy of the PDF, Figures 6 and 4 (p. 14) were overlapping and so really impossible to see.

Figure 1: In the text, the condition is referred to as "word list", but in the figure "wordlist". Having these match isn't strictly necessary but would be a nice touch.

Regarding accuracy differences: including accuracy in the models are good, but alternately, restricting analyses to only correct responses might also help with this?

*Reviewer #2 (Recommendations for the authors):*

– l. 122: does that mean the sentence was always presented first and the word-list second? The word-list did not contain the same words as the sentence, right? This might have affected differences in tracking and power.

– l. 197: why were the linear mixed-models for the MEG data performed separately per ROI? Also, were ROIs computed across hemispheres?

– l. 198: was the procedure described by Ince et al., 2017 using gaussian copula used for the MI analysis, if yes could you cite the reference?

– l. 191: could you give the details of the filter, what kind of filter etc?

– l. 191/l. 206: why were the frequency bands for the power analysis (δ: 0.5-3.0 Hz; theta: 3.0-8.0 Hz; α: 8.0-15.0 Hz; β: 15.0-25.0 Hz) different than for the mutual information analysis (l. 191: phrase (0.8-1.1 Hz), word (1.9-2.8 Hz), and syllable (3.5- 5.0 Hz)), could you explain the choice of frequency band? Particularly, if the authors want to check signal-to-noise differences in the mutual information analysis by using the power analysis, it seems relevant to match frequency bands.

– l. 207: why was the data averaged across trials and not the single-trial data fed into the mixed-models, such an analysis might strengthen the findings?

– l. 213: averaged across hemispheres or across trials?

– l. 216: all previous analyses were conducted across the hemispheres?

– l. 217: "for the four different frequency bands" this is referring to the frequency bands chosen for the power analysis, not those in the MI analysis?

– Figure 1: were only correct trials analyzed in the MEG analysis? If not, this might be a problem as there were more correct trials in the sentence compared to the word-list condition at the phrasal scale task. Could you add a control analysis on the correct trials only, to make sure that this was not a confound?

– l. 252 ff./Figure 2: As no power peak is observed in the δ (and α) band, this might result in spurious connectivity findings.

– l. 304: with higher connectivity in the phrasal compared to the passive task? Could you add this info?

– Something went wrong with figure 6.

– l. 403: alignment of neural oscillations with acoustics at phrasal scale, this reference seems relevant here: Rimmele, Poeppel, Ghitza, 2021.

– Wording: l. 387 "in STG".

*Reviewer #3 (Recommendations for the authors):*

1. In procedure settings, I am not sure whether the sentence condition is always presented before the word list condition (the misunderstanding comes from lines 126-128). Please add the necessary details and avoid this misunderstanding.

2. In the MEG analysis, PCA was performed to reduce computational load and increase the data relevance. However, the PCA procedure is not clear. For example, the authors extracted top *20 PCA components per region. Why 20 components? And why not 10 or 30 components? The details should be clarified.

---

## [Author Response]

Essential revisions:1. The relationship of participants' behavior to the observed MEG responses. One clear concern is the degree to which the accuracy differences between word lists and sentences (Figure 1) explain MI differences.2. A number of important details regarding analysis choices were missing and should be added, including: justification of the frequency bands used for the MI analysis, task order (were sentences always first, and word lists second), whether ROIs were defined across hemispheres, details on signal filtering, and details regarding the PCA analysis.3. The use of "hierarchical linguistic structure" should be clarified or tempered in the context of the current results. Although the attention was directed to syllables, words, and phrases through instructions, the significant MEG results were in the phrasal band (Figure 3) and not the others (supplemental figures).

We thank the Editor and the Reviewers for their helpful comments and constructive feedback. We feel that the reviews have substantially improved our manuscript. We have now addressed these three core concerns, and detail at length in individual responses to each reviewer how and what can now additionally be shown. To summarize briefly here:

1. We now include a new analysis in which we add accuracy as a factor. None of the original statistical patterns changed. We still find that MI/ neural tracking is higher for phrases in sentences than in word lists and that tracking during spoken language processing is a largely automatic response. Task specific effect were still found in in MTG and IFG. Thus, accuracy differences did not explain MI difference and we therefore remain with our core messages of the paper. We would like to note that we do no use tasks in the same way that they are typically used in cognitive neuroscience (say in a working memory paradigm, where only neural activity on correct trials can be associated the cognitive processing in question; instead we use tasks to direct our participant’s attention to syllables, words, and phrases and the timescale that they occur in). This difference in design – in other words, that in our case, that behavioral performance on the syllable task was lower than in the word or phrase tasks, does not mean that the sentence or word list was not heard/comprehended – means that inclusion or exclusion of incorrect trials does not further isolate the cognitive process of spoken language comprehension in question. That said, we agree it is very important to show that MI differences do not stem from behavioral performance differences. In order to show this, we add the above mentioned analysis to the manuscript. As said, this analysis did not change any of our main findings or conclusions, and rather strengthened the argument that tracking of phrases in sentences vs. word lists is stronger. We report now report these results in both our response and the manuscript.

2. We believe that we have fully addressed these queries and thank the reviewers for encouraging us be more comprehensive and thorough. We note that due to our 2-condition by 4-task design, we did not achieve enough power to include ROI as an independent factor. This is obviously not ideal, but embodies a trade-off between testing these 8 conditions together, which we felt was crucial for the inferences we wanted to make about tracking, and discovering underlying sources. We thus stuck to the well-known ROIs/ sources widely used and defined in the speech and language processing literature.

3. We regret any confusion this word might have caused; we meant to refer to the fact that our 4 tasks focus participants’ attention to different timescales and linguistic representations occur on across the linguistic hierarchy. We have changed the title to more clearly reflect this and have removed reference to linguistic hierarchy throughout the paper.

Reviewer #1 (Recommendations for the authors):The participants ranged in age from 18-59, which is a rather broad range. Given age-related changes in both hearing (not assessed) and language processing, it was surprising to not at least see age included in the models (though perhaps there were not enough participants?). Some comment would be useful here.

We indeed had quite a wide range of ages included in our study. Our wide age range was partly due to constraints in the MEG testing during covid times. It was difficult to find participants during the pandemic. On top of this, it is almost standard practice in the Netherlands to put dental wires behind the teeth after having braces and leave them in place for life. These participants cannot participate in MEG studies and therefore we widened the scope of our age range to reach enough participants. However, we also believe that it is good practice to not limit the participants age range to such a restricted range as is common practice in most cognitive neuroscience research. Otherwise, we end up only comparing brains of student populations with each other while we want to make conclusions for a much wider population. It is therefore rather an asset than a drawback that we have this wide age range. Regarding the hearing difficulties, we would like to note that we did ask participants whether they had any hearing problems (which none of them reported), but indeed did not assess did ourselves.

Nonetheless, we agree that it is valuable to investigate whether age had an influence in any of our results and therefore added age in the models of the main manuscript (i.e. for the lower frequencies). Again, we could only run fixed intercept models due to a reduction in the degrees of freedom.

For the MI we found no significant effect of age or interactions for any of our models. There was a trend significant effect in MTG for task*age interaction (F(3,123) = 2.5874, p = 0.0561). However, also the task*condition interaction remained significant and thus not change any conclusions.

**Author response image 1. sa2fig1:** Age effects on power estimates. (**A**) Predicted values for δ power for the two conditions dependent on age in STG (left) and MTG (right). (**B**) Predicted values for δ power for the four tasks dependent on age in MTG. Error bars indicate the 95% confidence interval of the fit. Colored lines at the bottom indicate individual datapoints.

For power we found a significant interaction between condition and age in STG (F(1, 28.88), p = 0.0192) and MTG (F(1,31) = 10.31, p = 0.003). In the MTG we additionally found an interaction between task and ages (F(3,31) = 5.02, p = 0.006). In IFG we only found a main effect of age (F(1,43) = 5.067, p = 0.030). For connectivity we found overall a main effect of age for all connections (STG-MTG, F(1,17.12) = 10.09, p = 0.005), (STG-IFG, F(1,16.96) = 17.42, p < 0.001, MTG-IFG, F(1,17.058) = 12.478, p = 0.002). The condition*age interaction in STG and MTG both suggested only for wordlist a change in power with age and not for the sentence condition (follow-up correlation age-MI per condition. STG: p = 0.076 (uncorrected) and MTG: p = 0.023 (uncorrected)). The task*age interaction in MTG showed only for the passive task a significant effect of age (follow-up correlation age-MI per task. p = 0.028 (uncorrected)).

Generally, we found it difficult to make strong conclusions about the main effect of age. Firstly, we did not have any baseline to assess whether main effects are specific to our language tasks or a more general age-related difference. Secondly, there could be anatomical age-related differences that spuriously drive the main effects of age. Therefore, we only report on the interaction effects. These analyses are now in the manuscript and there is an additional figure in the supplementary materials.

The results now read:

“Age control. Adding age to the analysis did not change any of the original findings (all original effects were still significant). We did however find for the power analysis age-specific interactions with condition and task. Specifically, for both the STG and the MTG we found an interaction between age and condition (F(1,28.87) = 6.156, p = 0.0192 and F(1,31) = 10.31, p = 0.003). In both ROIs there was a stronger difference between sentences and word lists (higher δ power for sentences) for the younger compared to the older participants (Supplementary Figure 4). In the MTG there was also an interaction between task and age (F(1,31) = 5.020, p = 0.006). Here, in a follow-up we found that only in the word task there was a correlation between age and power (p = 0.023 uncorrected), but not for the other tasks (p>0.1).”.

Figures: The y axis label seems frequently oddly positioned at the top – maybe consider just centering it on the axis?

We changed the y-axes of the figures accordingly and centered them all.

I liked the inset panels for the main effects, but they are rather small. If you can make them any larger that would improve readability.

We increased the size of the insets. Note however that as there is no space to add any y-labels these insets do not reflect real data, but just to quickly show the direction and presence of a main effect. We added this information now in the figure legend to ensure that this is clear.

In general, I really like showing individual subjects in addition to means. However, it also can make the plots a bit busy. I imagine you've tried other ways of plotting but might be worth exploring ways to highlight the means a bit more (or simplify the main paper figures, and keep these more detailed figures for supplemental)? Totally up to you, but I had a hard time seeing the trends as the plots currently stand.

There is a difficult balance to showing everything and highlighting the important bits. We were happy the reviewer pointed out that all trends were difficult to see and now change the figures to only include the mean and SEM. Additionally, when we found a main effect of task, we also added the condition means to highlight the differences on the significant effect more. The original plots are now added in the supplementary materials.

On my copy of the PDF, Figures 6 and 4 (p. 14) were overlapping and so really impossible to see.

We are so sorry about this! We realized this was the case due to the pdf conversion from our own word document. We now double-checked that this wasn’t the case.

Figure 1: In the text, the condition is referred to as "word list", but in the figure "wordlist". Having these match isn't strictly necessary but would be a nice touch.

We change the condition to word list throughout.

Regarding accuracy differences: including accuracy in the models are good, but alternately, restricting analyses to only correct responses might also help with this?

We have looked into this analysis, but there are some issues with doing the analysis in this way. First, there are clear differences in difficulty level of the trials within a condition. For example, if the target question was related to the last part of the audio fragment, the task was much easier than when it was at the beginning of the audio fragment. In the syllable task, if syllables also were (by chance) a part-word, the trial was also much easier. If we were to split up in correct and incorrect trials we would not really infer solely processes due to accurately processing the speech fragments, but also confounded the analysis by the individual difficulty level of the trials. Second, we end up with a very low trial amount after only looking at incorrect trials. In the worst case, some participants end up with only 22 trials which is too little to do much. We think it is therefore fair to compare the accuracy across participants (as done in the other analysis), but it is difficult to do this within participants.

To acknowledge this, we added this limitation to the methods. The method now reads:

“Note that different trials within a task were not matched for task difficulty. For example, in the syllable task syllables that make a word are much easier to recognize than syllables that do not make a word. Additionally, trials pertaining to the beginning of the sentence are more difficult than ones related to the end of the sentence due to recency effects.”

Reviewer #2 (Recommendations for the authors):– l. 122: does that mean the sentence was always presented first and the word-list second? The word-list did not contain the same words as the sentence, right? This might have affected differences in tracking and power.

We regret the confusion. We randomized whether the sentence or wordlist would come first across participants, but kept it constant within the participants across the four tasks. So, an individual would always alternate between a sentence and a word list block, but which one is first is counterbalanced across participants. To control for acoustic differences, we have the same words in the wordlists and the sentences (see e.g. Kaufeld et al., 2020). But this cannot have affected the results as the order is counterbalanced across participants. We now made clearer in the text how we assign the blocks.

– l. 197: why were the linear mixed-models for the MEG data performed separately per ROI? Also, were ROIs computed across hemispheres?

We had a clear a-priori idea about choosing these ROIs, therefore we did the analysis separately. Post-hoc, there now is also the simple issue of having too little power to estimate a three-way interaction between ROI, task, and condition. This power situation arises because we needed to test 4 task * 2 condition together in a single experiment in order to be able to make the inferences that we wanted to about the effects on tracking. This unfortunately comes at the cost of the power we can allot to each condition which affects our ability to generate source models with ROIs as a factor. Ideally, this would have been done, but practically this is very difficult to achieve. We cannot do more than acknowledge this in the main text (now in line 195-199). It is simply very difficult to have any design with multiple factors on top of an anatomical constraint (most studies seem to limit themselves to two factors these days to avoid these issues). The ROIs were computed across hemisphere (we had no a-priori hypothesis about hemispheres). Note that the PCAs therefore we also calculated across hemispheres (except for the coherence analysis as cross-hemisphere coherence seemed unlikely to us). Please note that we do not use ROIs that deviate from the established speech and language processing literature.

We now write:

“All following analyses were done per ROI. With enough statistical power one would add ROI as a separate factor in the analyses, but unfortunately, we did not have enough power to find a potential three-way interaction (ROI*condition*task). We therefore cannot make strong conclusions about one ROI having a stronger effect than another.”

– l. 198: was the procedure described by Ince et al., 2017 using gaussian copula used for the MI analysis, if yes could you cite the reference?

We regret not citing it in the main text and do this now.

– l. 191: could you give the details of the filter, what kind of filter etc?

We used a third order (bi-directional) Butterworth filter separately on eight equidistant bands of the speech signal. This information is now added.

– l. 191/l. 206: why were the frequency bands for the power analysis (δ: 0.5-3.0 Hz; theta: 3.0-8.0 Hz; α: 8.0-15.0 Hz; β: 15.0-25.0 Hz) different than for the mutual information analysis (l. 191: phrase (0.8-1.1 Hz), word (1.9-2.8 Hz), and syllable (3.5- 5.0 Hz)), could you explain the choice of frequency band? Particularly, if the authors want to check signal-to-noise differences in the mutual information analysis by using the power analysis, it seems relevant to match frequency bands.

The frequency bands of the MI analyses were based on the stimuli. They reflect the syllabic, word, and phrasal rates (calculated in Kaufeld et al., 2020). The power analyses were based on generic frequency bands. We choose for these two different splits as for the tracking one would expect the tracking (or phase alignment) from the exact frequency ranges in the signal relating to linguistic content (Ding et al., 2016; Martin, 2020; Keitel et al., 2018). In contrast, if there is no alignment, it is still possible that oscillatory signals are important in processing language stimuli (Jensen et al., 2010; Benitez-Burraco and Murphy, 2019). As these hypotheses rather pertain to commonly known frequency bands present in the brain we think it is more appropriate here to use generic bands (also to promote comparisons). We therefore stick to the stimulus-driven frequencies for the tracking hypothesis and use for control frequencies the generic ones (as we do not match them with anything in the stimulus). Note however, that when controlling for power in both the MI and the connectivity analysis, we did use the power of the respective band used for that specific analysis.

We write this reasoning now clearer in the methods:

“Power analysis was performed to compare the MI results with absolute power changes. On the one hand, we did this analysis as MI differences could be a consequence of signal-to-noise differences in the original data (which would be reflected in power effects). On the other hand, generic δ power has been associated with language processing [27, 28]. Therefore, we choose to analyse classical frequency bands instead of the stimulus informed ones (as used in the MI analysis) in order to compare these results with other studies. Moreover, as this analysis does not measure tracking to the stimulus (like the MI analysis does) it did not seem appropriate to match the frequency content to the stimulus content.”

– l. 207: why was the data averaged across trials and not the single-trial data fed into the mixed-models, such an analysis might strengthen the findings?

We could do this for the power analysis, but not for the MI and connectivity analysis which were calculated incorporating all trials (for the MI by concatenating all trials/speech signals, and for the connectivity as connectivity is calculated across trials). This would mean that we would run a different model on the power as on the other two variables of interest. We choose not to do this. Previously, MI analyses have been done by concatenating the data to improve sensitivity of the measure (Keitel et al., 2020; Kaufeld et al., 2018). Theoretically one could calculate the MI per trial, however this would unnecessarily reduce the sensitivity of the analysis.

– l. 213: averaged across hemispheres or across trials?

Across hemispheres. The WPLI is calculated across trials, so cannot be averaged across trials. We now clarified this.

– l. 216: all previous analyses were conducted across the hemispheres?

Yes. The MI and the power used the principles components incorporating both hemispheres, but for the connectivity analysis we calculated the PCAs for the hemispheres separately and averages only after calculating the connectivity across hemispheres.

– l. 217: "for the four different frequency bands" this is referring to the frequency bands chosen for the power analysis, not those in the MI analysis?

Yes. This is now clarified. As stated above there it makes more sense to look at generic frequency bands when we do not directly link the analysis with the speech signal.

– Figure 1: were only correct trials analyzed in the MEG analysis? If not, this might be a problem as there were more correct trials in the sentence compared to the word-list condition at the phrasal scale task. Could you add a control analysis on the correct trials only, to make sure that this was not a confound?

We included all trials in the analysis; we do this for several reasons, first because we not use the tasks in the same way as a traditional working memory task, where only correct trials contain the cognitive process that is being studied. Second, because it is not possible to only include the correct trials as we end up with too few trials per condition to reliably estimate our effect. Moreover, trials are not controlled for difficulty within a condition. However, in order to show that our MI effects are not driven by task performance, we can include accuracy across participants as a factor. We refer to point 1 for the results of this analysis. We have added this information now in the methods:

“Note that different trials within a task were not matched for task difficulty. For example, in the syllable task syllables that make a word are much easier to recognize than syllables that do not make a word. Additionally, trials pertaining to the beginning of the sentence are more difficult than ones related to the end of the sentence due to recency effects.”

– l. 252 ff./Figure 2: As no power peak is observed in the δ (and α) band, this might result in spurious connectivity findings.

Indeed, the absence of a peak does result in difficulty to interpret connectivity findings as it is unknown whether we are truly investigating an endogenous oscillation or some other form of connections which could for example be related to stimulus evoked responses. If the latter, we could not speak of ‘true’ connectivity increases, but rather for differential processing of the stimulus. We acknowledge this, but still find also this type of change worth reporting. We therefore made clear cautious note in the Results section that this is the case.

The results now read:

“Note that not for all frequency bands we found a clear peak in the power signal (only clearly so for the α band), this indicates that the connectivity likely does not reflect endogenous oscillatory activity [29], but might still pertain to connected regions operating at those time scales.”

The discussion now reads:

“Note that even though we found increased connectivity, we did not see a clear power peak in the δ band. This suggest that we might not be looking at an endogenous oscillator, but rather at connections operation at that temporal scale (potentially being non-oscillatory in nature).”

– l. 304: with higher connectivity in the phrasal compared to the passive task? Could you add this info?

We have added this info.

– Something went wrong with figure 6.

We have realized this and regret it. The new manuscript should have the figures placed correctly.

– l. 403: alignment of neural oscillations with acoustics at phrasal scale, this reference seems relevant here: Rimmele, Poeppel, Ghitza, 2021.

Agreed at we have added this.

– Wording: l. 387 "in STG".

We have changed this wording accordingly.

Reviewer #3 (Recommendations for the authors):1. In procedure settings, I am not sure whether the sentence condition is always presented before the word list condition (the misunderstanding comes from lines 126-128). Please add the necessary details and avoid this misunderstanding.

We regret the confusion. We randomized whether the sentence or wordlist would come first across participants, but kept it constant within the participants across the four tasks. So, an individual would always alternate between a sentence and a word list block, but which one is first is counterbalanced across participants. To control for acoustic differences, we have the same words in the wordlists and the sentences (see e.g. Kaufeld et al., 2020). But this cannot have affected the results as the order is counterbalanced across participants. We now made clearer in the text how we assign the blocks.

2. In the MEG analysis, PCA was performed to reduce computational load and increase the data relevance. However, the PCA procedure is not clear. For example, the authors extracted top *20 PCA components per region. Why 20 components? And why not 10 or 30 components? The details should be clarified.

As the reviewer also mentions, our main aim was to reduce the dimensions of our analysis (to reduce computational load), while keeping in enough variance relevant for the analysis. We are not aware of any established way to determine the amount of PCA components which is best to use and can therefore only look at the explained variance of the components. As for us the goal was to reduce computational load, we wanted to keep as much original variance in the analysis as possible. In Author response image 2 one can see the cumulative explained variance of the components. Around 20 components over 99.9 % of the data is explained, which seems sensible amount to keep in.

To ensure that all our results are also robust against changing the exact number of components we repeated the analysis using 10 and 30 components. No big qualitative differences were visible. It did seem that 10 components were not sufficient to show the original effects. The IFG task effect and MTG task*condition effect were only trend significant for 10 components (p = 0.06 and p = 0.1 respectively). The condition effect remained significant for all ROIs. Using 30 components all effects (including the main and interaction; IFG task effect: p = 0.034; MTG task*condition interaction p = 0.0064) effect remained significant. We therefore believe that the results are robust against choosing the exact amount of PCAs, but does require more than 10 components.

We now discuss the effect of PCA component number in the text. The text now reads:“Overall, the amount of PCA components did not influence any of the qualitative differences in the condition. It did seem however that 10 PCA components were not sufficient to show all original effects with the same power. Specifically, the IFG task and MTG task*condition effect were only trend significant for 10 components (p = 0.06 and p = 0.1 respectively). The other effects did remain significant with 10 components. Using 30 components made some of our effects stronger than with 20 components. Here, the IFG task and MTG task*condition effects had p-values of 0.034 and 0.006 respectively. We conclude that the amount of PCAs components did not qualitative change any of our reported effects.”